# Cortisol and Dexamethasone Mediate Glucocorticoid Actions in the Lesser Spotted Catshark (*Scyliorhinus canicula*)

**DOI:** 10.3390/biology11010056

**Published:** 2021-12-31

**Authors:** Juncal Cabrera-Busto, Juan M. Mancera, Ignacio Ruiz-Jarabo

**Affiliations:** 1Departament of Biology, Faculty of Marine and Environmental Sciences, Campus de Excelencia Internacional del Mar (CEI-MAR), Universidad de Cádiz, 11510 Puerto Real, Spain; juncal.cabrera@uca.es (J.C.-B.); juanmiguel.mancera@uca.es (J.M.M.); 2Department of Physiology, Faculty of Biological Sciences, University Complutense Madrid, 28040 Madrid, Spain

**Keywords:** cortisol, dexamethasone, glucocorticoid receptors, *Scyliorhinus canicula*, sharks

## Abstract

**Simple Summary:**

For the first time, glucocorticoid actions of corticosteroids are evidenced in vivo and ex vivo in sharks, highlighting the importance of carbohydrate metabolism in situations of high-energy expenditure in this taxonomical group. Long-term (7 days) in vivo administration of dexamethasone (DEX, a synthetic glucocorticoid) decreased 1α-hydroxycorticosterone (1α-OHB, the main corticosteroid hormone in sharks), while also modified carbohydrates metabolism in liver and white muscle. Short-term (1 to 5 h) ex vivo incubation of liver and muscle explants with cortisol (corticosteroid not present in sharks) and DEX revealed glucose secretion mediated by glucocorticoid receptors (GR), as seen by the employment of mifepristone (a GR inhibitor).

**Abstract:**

Corticosteroids are hormones produced in vertebrates exerting gluco- and mineralocorticoid actions (GC and MC) mediated by specific receptors (GR and MR, respectively). In elasmobranchs, the major circulating corticosteroid is the 1α-hydroxycorticosterone (1α-OHB). This hormone acts as a MC, but to date its role as a GC has not been established. As there is no 1α-OHB standard available, here we employed a set of in vivo and ex vivo approaches to test GC actions of other corticosteroids in the lesser spotted catshark (*Scyliorhinus canicula*). Dexamethasone (DEX, a synthetic corticosteroid) slow-release implants decreased plasma 1α-OHB levels after 7 days, and modified carbohydrates metabolism in liver and white muscle (energy stores and metabolic enzymes). In addition, ex vivo culture of liver and white muscle explants confirmed GC actions of corticosteroids not naturally present in sharks (cortisol and DEX) by increasing glucose secretion from these tissues. Dose–response curves induced by cortisol and DEX, altogether with the use of specific GR inhibitor mifepristone, confirmed the involvement of GR mediating glucose secretion. This study highlights the influence of corticosteroids in the glucose balance of *S. canicula*, though the role of 1α-OHB as a GC hormone in sharks should be further confirmed.

## 1. Introduction

Corticosteroids are involved in the endocrine regulation of several physiological processes such as metabolism, growth, hydromineral balance, and the immune system [1]. In early vertebrates, these steroid hormones are produced by interrenal cells and secreted into the bloodstream, and the major corticosteroid is taxon dependent: cortisol in teleosts and Acipenseriformes [2,3], 11-deoxycortisol in lampreys [4], and 1α-hydroxycorticosterone (or 1α-OHB) in elasmobranchs [5].

Steroids are cholesterol derivatives maturated through specific enzymes [6], and their actions induce a response through binding to specific receptors [7,8]. Depending on their function, there are glucocorticoid (GC) and mineralocorticoid (MC) receptors [9,10]. While Agnatha show only one corticosteroid receptor, separated GC and MC receptors first appear in Chondrichthyes (sharks, rays, and skates) [11,12]. Glucocorticoid receptors (GR) are involved in several physiological processes, including regulation of carbohydrate metabolism [13], while MC receptors (MR) regulate hydromineral balance [8].

Corticosteroids have been widely studied in some vertebrate groups, like cyclostomes [4] and teleosts [14]. Interestingly, in these vertebrate groups, corticosteroid hormones have a dual role as GC and MC. The appearance of differentiated GC and MC hormones, mostly cortisol/corticosterone and aldosterone only happened after the evolution of Dipnoi [15], and this division is maintained in terrestrial vertebrates. However, there are some gaps in the knowledge of this topic in other groups, such as elasmobranchs.

In this sense, 1α-OHB as the predominant corticosteroid in sharks shows MC activity [16]. In sharks and Dipnoi, MC action of corticosteroids was studied by the employment of dexamethasone, a synthetic analogue of cortisol [17,18]. Recent studies evidenced a possible role of 1α-OHB as a GC in sharks after an acute-stress challenge in the lesser spotted catshark (*Scyliorhinus canicula*). In that study, plasma 1α-OHB and glucose enhanced after air exposure showing a linear relationship [19], similarly to what was described for cortisol and glucose in teleosts [20]. However, other authors cautiously addressed that further studies are mandatory to assert a possible GC role of corticosteroids in sharks [21].

The predominant GCs produced in vertebrates—cortisol and corticosterone—are pleiotropics [2], modeling a wide range of physiological processes, including energy metabolism, through the stimulation of hepatic gluconeogenesis and proteolytic processes in the muscle as well as lipolysis in the adipose tissue to increase glucose levels. Presumably, GC responses in sharks may differ from other taxa due to their unique intermediary metabolism, although carbohydrates appear to be utilized similarly in sharks to other vertebrates [22], offering the possibility of studying GC actions through the analysis of glucose metabolism. As there are evolutionary similarities between sharks and teleosts, it could be (cautiously) assumed that long and short term GC responses may be different, as described in teleosts [23,24]. Thus, shark studies can be benefited from the vast amount of information describing the effects of GC in teleost fish [1,25]. However, only select lessons from these studies can be confidently applied to elasmobranchs [26].

In this sense, the teleost *Sparus aurata* enhanced, during chronic administration of cortisol and dexamethasone, the general catabolism at the cost of compromising its body growth [23]. However, these authors described great differences in carbohydrate metabolism responses depending on the target tissue. Thus, the liver in fish act as a key organ of energy storage, mostly in the form of glycogen and triglycerides [1,27], while muscle has glycogen, which can be required to fuel glycolysis [28]. Furthermore, it was described that corticosteroids stimulate hepatic gluconeogenic pathways and consumption of white muscle glycogen stores in vertebrates [29,30,31].

Glucose metabolism also results of major importance after short-term situations as described in vertebrates, including sharks, to provide energy to demanding tissues [32,33,34]. Acute responses of cortisol include rapid glycolytic actions mediated by GR in teleost fish that affect liver and muscle [35,36]. Although GC actions in vertebrates from lampreys to mammals are well described, there is a knowledge gap on how these hormones affect sharks in the long and short term [37].

Thus, the aim of this study was to deepen further into the GC actions of corticosteroid hormones in sharks, using *S. canicula* as a biological model. This evaluation is *included* in vivo and ex vivo experiments where the effects of dexamethasone and cortisol were tested.

## 2. Materials and Methods

### 2.1. Ethics Statement

This study was performed in accordance with the Guidelines of the European Union (2010/63/UE) and the Spanish legislation (RD 1201/2005 and law 32/2007) for the use of laboratory animals. This study did not involve endangered or protected species. All experiments have been carried out under a special permit granted to the Spanish Institute of Oceanography and approved by the Spanish General Secretariat of Fisheries (project DISCARDLIFE, Fundación Biodiversidad, Ministry for the Ecological Transition, Spain).

### 2.2. Animal Maintenance

Lesser spotted catshark (*S. canicula*) adults of both sexes (n = 22, 345.1 ± 58.82 g body weight and 50.7 ± 2.67 cm total length, mean ± SD.) were obtained by bottom trawling as described before [38] and maintained in the fish husbandry facility of the Faculty of Marine and Environmental Sciences (Puerto Real, Cadiz, Spain) until the beginning of the experiment. All maintenance procedures were described in previous studies [19,39]. Fish were randomly divided into 6 tanks of 400-L (surface area of 0.72 m^2^, 0.56 m depth, and covered by overshadowing tissue) at a stocking density previously described for this and other demersal shark species [19,39,40,41]. The system consisted in a flow-through supply of seawater (38 psu), natural photoperiod (March to June; latitude 36°31′34″ N) and temperature (ambient temperature of 19 ± 0.3 °C) along the acclimation period (>23 days). Fish were fed every other day at 12:00 UTC with fresh shrimps, prawns, sardines, and anchovies to satiety. Animals were fasted 36 h before sampling in order to avoid blood imbalances (acid–base balance, energy metabolites, ions) related to feeding, as was described before in dogfish [42].

### 2.3. Long-Term In Vivo Responses

Due to the absence of purified 1α-OHB, alternatives have been sought that may provide some information on the effects of corticosteroids on this species. As it was described that dexamethasone (a synthetic analogue of cortisol) exerts MC actions in *S. canicula* [17], we decided to employ this drug to assess GC actions in this species. Catsharks (n = 7 per group) were anesthetized with 2-phenoxyethanol (0.05% *v*/*v*), weighed and randomly received an intraperitoneal (IP) injection with slow-release coconut oil alone (4 females and 3 males, 5 µL oil g^−1^ body weight, serving as the control group, C-1758 Sigma-Aldrich, St. Louis, MO, USA), or containing dexamethasone (3 females and 4 males, 0.103 µmol g^−1^ body weight, D-175650, Sigma-Aldrich). The amount of implanted dexamethasone was based on a previous study carried out in *S. canicula*, where daily injections were administered for 4 days [17], but also on a study carried out in the teleost *S. aurata* treated with slow-release cortisol implants [31]. This treatment is effective in maintaining high levels of implanted steroid hormones in fish blood for more than 7 days [31,43,44]. After injection, fish were transferred to the same experimental tanks and maintained for 7 days in the same conditions described above, being fed every other day and fasted 36 h before sampling.

### 2.4. Acute Ex Vivo Responses

Liver and white muscle (from the dorsal part behind the first dorsal fin) explants were ex vivo incubated based on previous protocols developed for teleost pituitary glands [45], and rat, goat, and human liver slices [46,47]. Glucose secretion was evaluated as a GC-mediated response. The incubation medium was developed ad hoc for *S. canicula* after the analysis of plasma from control-undisturbed animals (Appendix A) maintained in similar conditions as those in the present study [39]. The incubation medium contained 208.6 mM Na^+^, 284 mM Cl^−^, 2.9 mM K^+^, 5.3 mM Ca^2+^, 2 mM Mg^2+^, 3.6 mM PO_4_^−^, 2 mM SO_4_^2−^, 3.4 mM HCO_3_^−^, 385 mM urea, a variable amount of glucose depending on the experiment (see below); and it was supplemented with 10 μL mL^−1^ vitamins (MEM 100× Vitamins, M-6895, Sigma–Aldrich), 20 μL mL^−1^ essential amino acids (MEM 50×, M-5550, Sigma–Aldrich), 10 μL mL^−1^ non-essential amino acids (MEM 100×, M-7145, Sigma–Aldrich), 10 μL mL^−1^ antibiotics (penicilin 10,000 IU mL^−1^; streptomycin 10 mg mL^−1^, P-0781, Sigma–Aldrich), and 20 μL mL^−1^ L-glutamine (200 mM, G-7513, Sigma–Aldrich); adjusted to pH 7.8 with Trizma Base (T6791, Sigma-Aldrich) and osmolality of 925 mOsm kg^−1^.

Once the tissues were collected and cut into cubic pieces of the appropriate size (<5 mm long side), explants were individually maintained in 1 mL plasma-like incubation buffer (as described above) for 60 min to accommodate fish needs, allowing the tissue to recover after the feasible stress induced by dissection. After this time, explants were individually weighted and transferred to 24-well microplates containing 1 mL fresh incubation medium and incubation time started. All procedures were conducted at 25 °C. Glucose secretion rates were analyzed in liver and muscle explants ranging from 0.05 to 0.20 g weight in control-undisturbed animals under different conditions (described under these lines). Glucose secretion rates were calculated as the difference between the final and the initial glucose concentration in the incubation medium, divided by the tissue explant mass and the incubation time.

The effect of cortisol (H-4001, Sigma-Aldrich), and dexamethasone as a synthetic analogue of cortisol (D-1756, Sigma-Aldrich), as well as a glucocorticoid receptor inhibitor (RU-486 or mifepristone; M-8046, Sigma-Aldrich) were tested in the incubations. Although RU-486 also inhibits progesterone receptors [48], to the best of our knowledge, these receptors do not mediate glucocorticoid actions [49]. Cortisol and dexamethasone were tested at 3 nM which is the maximum concentration of 1α-OHB in *S. canicula* after 18 min air-exposure [19]. Both hormones activate in vitro transcription of GR in elephant shark and the African lungfish, while RU-486 inhibits elephant shark GR [18,50]. Moreover, as a proxy to evaluate pharmacological consequences of these molecules, higher concentrations were also tested (30 nM, 0.3 µM, 3 µM, and 30 µM) in two females, covering the range described of physiological concentrations of cortisol in teleost fish [20,51,52]. This range was tested before in other ex vivo incubation studies in teleost fish [53,54,55], and it is assumed to be above the physiological concentrations of corticosteroids in sharks. RU-486 was tested at 30 µM, which is the concentration with maximum inhibition of glucocorticoid receptors in mammals and teleost fish [53,54,56,57,58]. All incubations, including the control groups, were performed with 0.006% DMSO (dimethylsulfoxide; D-8779, Sigma-Aldrich), since DMSO was used to pre-dissolve these compounds.

A time course survey was performed for the explants ex vivo incubated, in the same conditions described before, to test the glucose secretion rate for 1, 2, and 5 h, coinciding with the described maximum rates of glucose secretion in *S. canicula* and teleost fish [19,24].

Different concentrations of glucose in the incubation medium were also tested, including 0, 2.3 (plasma basal concentration observed in *S. canicula*) and 4.6 mM (maximum plasma concentration described in *S. canicula* 5 h after an air-exposure challenge) [19].

All the experimental assays were performed together with a control group, using all tissue explants from a single animal to test experimental vs. control condition *ceteris paribus*. Repeated independent experiments with at least 2 samples for each experimental condition and animal were conducted in order to ensure proper statistical robustness of the results. A preliminary test was conducted with only 2 animals (females) to assess glucose secretion due to cortisol/dexamethasone concentration, which was evaluated with 8 explants per animal. The time course of glucose secretion, the effect of initial glucose concentration and the use of a glucocorticoid receptor inhibitor were evaluated in 6 catsharks (3 females and 3 males), using 2 explants per animal.

### 2.5. Sampling

All fish were sampled between 9.00 UTC and 11.00 UTC. Before sampling, catsharks were captured by hand and immediately anesthetized in 0.1% *v/v* 2-phenoxiethanol (P-1126, Sigma-Aldrich). Weight and length of the animals were recorded. Blood collected with heparinized syringes from the caudal vessels and placed into heparinized tubes. Plasma was separated from cells by centrifugation of whole blood (3 min, 10,000× *g*, 4 °C) and snap frozen in liquid nitrogen. Euthanization was confirmed by severing the head with a sharp knife. All procedures lasted less than 4 min per tank in order to obtain blood and tissue samples that reflect the condition prior to capture, as described for teleosts [59]. Liver was collected and weighed for hepatosomatic index (HSI) determination. Liver and white muscle samples were immediately frozen in liquid nitrogen. Samples were stored at −80 °C. White muscle collected from the dorsal part below the first dorsal fin was also collected to analyze muscle water content. Liver and white muscle samples for ex vivo incubations were collected and immediately placed in incubation medium, as described above.

### 2.6. Plasma Hormones Determination

To determine the concentration of steroids, an internal standard (1.93 µL of 100 µg/mL cortisol-d_4_, C-113, Sigma-Aldrich) is added to each plasma sample (1 mL). The extraction process was conducted twice in 750 µL methanol by vortexing, allowing it to settle for 15 min, followed by centrifugation at 10,500× *g* for 30 min at 4 °C. The supernatants from each centrifugation process were collected and pooled into another vial. The total volume of the combined supernatants was vortexed and centrifuged at 10,500× *g* for 30 min at 4 °C. The final supernatant was transferred to 10 mL tube and 8 mL of ultrapure water was added. After conditioning a C18 SPE column with 3 mL of methanol followed by 3 mL of ultrapure water, the sample was loaded. The column was washed with 4.5 mL H_2_O/MetOH (65:35; *v*/*v*) and retained compounds were eluted with 2 mL H_2_O/MetOH (20:80; *v*/*v*) into a 2 mL Eppendorf and evaporated to dryness at 37 °C. The sample was finally reconstituted in 200 µL MetOH. These procedures were based on a previous study [19] with modifications.

Chromatographic analysis was performed on a UPLC-QTOF-MS (Xevo-G2-S, Waters, Milford, CT, USA) using Acquity Ultra Performance BEH C18 (1.7 µm; 2.1 mm × 50 mm) column (Waters, Milford, CT, USA) from the Central Research Services (University of Cadiz, Cadiz, Spain). Glucocorticoids were separated using a gradient elution of mobile phases A and B. Mobile phase A was a mixture of ultrapure water with 0.1% formic acid, while phase B was a mixture of methanol with 0.1% formic acid. Initially, gradient elution started at 30% (*v*/*v*) of phase B. Subsequently, mobile phase was increased to 80% at 3 min, preserved in this way to 1 min and finally decreased to 30%. The flow rate was maintained constant at 0.4 mL min^−1^, resulting in a 5 min running time. The injection volume was set at 10 µL. The column temperature was maintained at 40 °C. The mass spectrometer was used in negative electrospray ionization mode (ESI^-^). The MS detector settings were: source temperature 120 °C, desolvation temperature 450 °C at a gas flow of 1000 L h^−1^, cone gas flow 10 L h^−1^, and capillary voltage 3 Kv. Calibration curves were prepared with standards of dexamethasone (D-1756, Sigma-Aldrich) and cortisol (H-4001, Sigma-Aldrich), dissolved in methanol. Subsequently, the stock factor was calculated and results were corrected. Since there is no commercial standard for 1α-OHB, but taking into account the similarities in its chemical structure and the same molecular weight as cortisol, which is absent in elasmobranchs [12], cortisol was used as standard for 1α-OHB in the UPLC-QTOF-MS as done before [19]. The retention time (min) and the precursor ion (m/z) obtained to dexamethasone, 1α-OHB, cortisol, and cortisol-d^4^ were 2.1, 1.5, 1.55, and 1.55 min, and 437.19, 407.20, 437.19, and 411.24 m/z, respectively. The limits of detection of the UPLC-QTOF-MS were found above 1379.46 µM and below 0.25 nM, calculated for cortisol. The intraspecific variation between samples were 0.05 ± 0.10 nM for 1α-OHB and 0.34 ± 0.22 f µM for dexamethasone.

### 2.7. Tissue Metabolite Content

Frozen liver and muscle were finely minced on an ice-cooled Petri dish and divided into two aliquots to assess enzyme activities and metabolite levels. The frozen tissue used for the assay of metabolites was homogenized by ultrasonic disruption in 7.5 volumes ice-cold 0.6 N perchloric acid, neutralized using 1 M potassium bicarbonate, centrifuged (3 min at 10,000× *g*, Eppendorf 5415R), and the supernatant used to assay (i) lactate (Lactate ref. 1001330, Spinreact SA, Sant Esteve de Bas, Spain); (ii) glycogen [60]; and (iii) glucose obtained after glycogen breakdown (after subtraction of free glucose levels) was analyzed with a commercial kit (Glucose-HK ref. 1001200, Spinreact SA, Sant Esteve de Bas, Spain). Muscle water content was analyzed by drying pre-weighted muscle at 65 °C until constant weight (circa 72 h). The percentage of water was calculated as the difference in weight between the fresh and the dry muscle divided by the fresh weight.

### 2.8. Enzyme Activities

Aliquots of liver and muscle were homogenized by ultrasonic disruption in 10 volumes of ice-cold stop buffer (250 mM sucrose, 50 mM imidazol, pH 7.5, 1 mM 2-mercaptoethanol, 50 mM NaF, 4 mM EDTA, 0.5 mM PMSF, and a protease inhibitor cocktail (Sigma, P-2714)). The homogenate was centrifuged (10 min at 9000× *g*, 4 °C) and the supernatant used in enzyme assays for the analysis of (i) glycolytic-related enzymes including HK (hexokinase, EC 2.7.1.1), PK (pyruvate kinase, EC 2.7.1.40), and GP (glycogen phosphorylase, EC 2.4.1.1); and (ii) gluconeogenesis-related enzymes (fructose 1,6-bisphosphatase-FBP, EC 3.1.3.11; lactate dehydrogenase-LDH, EC 1.1.1.27) as described before in *S. canicula* (Ruiz-Jarabo et al., 2019). The reactions were started by addition of 15 μL homogenate at a pre-established protein concentration, omitting the substrate in control wells (final volume 275–295 μL). Data were expressed as U mg^−1^ prot.

### 2.9. Plasma Metabolites and pH

Plasma glucose and lactate levels were measured using commercial kits from Spinreact (as described above) adapted for 96-well microplates. All assays were performed using a Bio-Tek PowerWave 340 Microplate spectrophotometer (Bio-Tek Instruments, Winooski, VT, USA) using KCjunior Data Analysis Software for Microsoft Windows XP unless otherwise stated. Plasma pH was measured immediately after centrifugation with a mini-electrode (HI1083B, Hanna Instruments, RI, USA).

### 2.10. Statistics

Normality and homogeneity of variances were analyzed using the Shapiro–Wilk’s test and the Levene´s test, respectively. Differences between groups for the long-term in vivo response to dexamethasone IP-implants were tested by a Student´s *t*-test. Differences between ex vivo incubation times were tested using one-way ANOVA with time (1, 2, and 5 h) as the factors of variance. Differences due to the ex vivo incubation with different cortisol and dexamethasone concentrations were tested using two-way ANOVA with hormone concentration (0, 0.003, 0.03, 0.3, 3, and 30 µM) and hormone (cortisol and dexamethasone) as the factors of variance. Differences in the ex vivo incubations with different initial concentrations of glucose were tested using two-way ANOVA with group (control, cortisol, and dexamethasone) and glucose (0, 2.3, and 4.6 mM) as the factors of variance. Differences due to the use of a glucocorticoid receptor inhibitor were tested using two-way ANOVA with group (control, cortisol, and dexamethasone) and inhibitor (control and RU-486) as the factors of variance. When necessary, data were logarithmically transformed to fulfill the requirements of ANOVA. When ANOVA yielded significant differences, Tukey´s post-hoc test was used to identify significantly different groups. Sex was included as a covariable in the in vivo and ex vivo experiments to analyze possible differences due to it. Statistical significance was accepted at *p* < 0.05. All the results are given as mean ± SD.

## 3. Results

No mortality occurred during the experiments. No differences due to sex of the animals were observed in any of the conducted experiments (data not shown).

### 3.1. Long-Term In Vivo Effects of a Corticosteroid IP-Implant

#### 3.1.1. Plasma Hormones

Dexamethasone was not detected in plasma from the control group, while those catsharks with slow-release IP-implants of this synthetic corticosteroid showed 3.40 ± 2.17 µM dexamethasone after 7 days. Consequently, IP implants of dexamethasone were successful, resulting in the release and transport of this hormone through the bloodstream.

This treatment also altered plasma values of the natural corticosteroid hormone in sharks, 1α-OHB. Plasma 1α-OHB concentration decreased by half in the dexamethasone-treated group (0.52 ± 0.41 nM) compared to the control group 1.06 ± 0.32 nM (Figure 1). As expected, no cortisol was detected in plasma of *S. canicula* from both groups.

#### 3.1.2. Tissue Metabolites and Enzyme Activities

In plasma, no significant differences were observed in hematocrit, pH, glucose, or lactate (Table 1).

Dexamethasone IP-implants induced a glycogen increase of 58.4% in the liver. In addition, a 24.9% decrease on hepatic GP activity, which converts glycogen into free glucose, was also observed (Figure 2). However, significant differences were detected neither in free glucose or lactate concentration nor in HK, PK, LDH, and FBP enzymatic activities in this tissue. No differences were observed either in the HSI (Table 1).

In white muscle, IP-implants of dexamethasone produced a glycogen depletion of 69.6%, accompanied by a 14.2% increase on GP activity (Figure 2). This treatment also significantly modified the enzymes HK (the first enzyme of the glycolysis pathway increased a 99.8% its activity), PK (the last enzyme of the glycolysis pathway enhanced its activity by 52.4%), and LDH (enzyme involved in the gluconeogenesis pathway stimulated up to 61.3% the activity of the control group) activities (Table 1). However, no changes were detected in FBP activity or in glucose and lactate concentrations. Finally, muscle water content showed an increase of 5.9% in dexamethasone-treated fish compared to the control group (Table 1).

### 3.2. Short-Term Ex Vivo Glucocorticoid Actions of Corticosteroids

#### 3.2.1. Incubation Time

Glucose secretion rates were constant in control groups along the 5 h tested (averaging 1.91 ± 0.81 µmol g^−1^ h^−1^ and 1.30 ± 0.52 µmol g^−1^ h^−1^ for liver and muscle explants, respectively). The highest glucose secretion rates were measured after 1 h of ex vivo incubation with both glucocorticoids tested (3 nM cortisol or dexamethasone) in both tissues analyzed (liver and white muscle). No differences were described between both corticosteroids tested at each sampling time. Thus, hepatic glucose secretion rates with cortisol and dexamethasone were 4.77 ± 1.34 and 5.41 ± 1.38 µmol g^−1^ h^−1^ after 1 h incubation, and decreased to 1.37 ± 0.55 and 1.63 ± 0.73 µmol g^−1^ h^−1^ after 5 h incubation. A similar tendency was described in white muscle, while the rates were lower than in liver. In this sense, glucose secretion rates with cortisol and dexamethasone were 3.06 ± 1.25 and 2.70 ± 1.18 µmol g^−1^ h^−1^ after 1 h incubation, and decreased to 1.14 ± 0.69 and 1.34 ± 0.47 µmol g^−1^ h^−1^ after 5 h incubation (Figure 3).

#### 3.2.2. Effect of Corticosteroids Concentration

A range of cortisol and dexamethasone concentrations (above the physiological concentrations of 1α-OHB in this species) was tested and the effects on glucose secretion (after 1 h incubation with an initial concentration of glucose in the incubation medium of 2.3 mM) in liver and white muscle was assessed (Figure 4). The minimum concentration of both hormones (3 nM, which may be related to the described levels of 1αOHB in this species after an air-exposure challenge) [19], significantly increased glucose secretion in both tissues, with no differences between hormones. Thus, 3 nM of cortisol or dexamethasone enhanced glucose secretion > 2.63-fold in liver and > 1.81-fold in white muscle. Higher concentrations of these corticosteroids (up to 30 µM) in liver did not change glucose secretion rates compared to the minimum concentration tested (3 nM). However, white muscle incubated explants decreased glucose secretion rates at 30 nM hormones, maintaining the maximum secretion rates in all other tested concentrations.

#### 3.2.3. Effect of Initial Glucose Concentration

To test the ex vivo glucose secretion rates by the tissues (liver and muscle), we considered it necessary to test the effects of the initial glucose concentration in the incubation medium (Figure 5). Only liver was tested herein due to the observed higher glucose secretion rates compared to white muscle. Maximum rates are described at physiological concentrations of control-undisturbed catsharks (2.3 mM glucose) [19], with 5.02 to 5.28 µmol g^−1^ h^−1^ in 3 nM cortisol- and dexamethasone-treated liver explants. However, physiological concentration of glucose similar to that observed in acute-stressed catsharks (4.6 mM) [19] promoted lower secretion rates (2.88 to 3.77 µmol g^−1^ h^−1^ when liver explants were incubated with 3 nM cortisol or dexamethasone). Cortisol or dexamethasone, in the absence of initial concentration of glucose in the ex vivo incubation medium, managed to increase glucose secretion rates compared to the control group, but the rates are significantly lower than those observed in the presence of glucose levels similar to physiological concentrations of control-undisturbed catsharks (reaching 2.15 to 2.18 µmol glucose g^−1^ h^−1^ when explants were incubated with 3 nM cortisol or dexamethasone).

#### 3.2.4. Glucocorticoid Receptors

The possible action of glucocorticoid receptors on glucose secretion induced by corticosteroids (cortisol and dexamethasone) has been tested through the use of a specific inhibitor, 30 µM RU-486 (Figure 6). No effect of RU-486 on the basal release of glucose was observed. Both cortisol and dexamethasone (3 nM), enhanced glucose secretion rates from 1.23 to 5.1 µmol g^−1^ h^−1^ in hepatic ex vivo explants incubated at an initial glucose concentration of 2.3 mM for 1 h. However, in the presence of corticosteroids and RU-486, these rates decreased to basal levels (control) of circa 2.00 µmol g^−1^ h^−1^, without differences compared to the control group incubated with RU-486 and no hormones (1.63 ± 0.77 µmol g^−1^ h^−1^).

## 4. Discussion

For the first time, there are evidences showing that corticosteroids in the lesser spotted catshark induced GC actions, including mobilization of glucose from liver and muscle. Moreover, GR mediated this glucose secretion, confirming current knowledge in other vertebrates. By using holistic in vivo and ex vivo approaches, this study highlighted the importance of carbohydrate metabolism in sharks during long- and short-term situations with high energetic demands.

### 4.1. Effect of Long-Term In Vivo Dexamethasone Treatment in Plasma Corticosteroids

Slow-release implants of the synthetic corticosteroid dexamethasone successfully managed to circulate this hormone through the bloodstream in *S. canicula*, as seen by its presence in plasma after 7 days injection. Previous studies employing this methodology tested the effect of corticosteroids and other hormones in teleost fish [31,43,44]. In all cases, validation of the experiments required the analysis of the targeted hormones in the blood. Due to the lack of purified standards of 1α-OHB, the techniques to measure its concentration accurately are limited and required other steroid hormones that may act as secondary standards [19]. In this sense, the use of liquid chromatography coupled to mass spectrometry (UPLC-QTOF-MS) for the analysis of steroids has revealed as one of the recommended methodologies due to the improved specificity compared to other techniques (like immunoassays), and the ability to multiplex panels of analytes, amongst other advantages [61,62]. Thus, through UPLC-QTOF-MS, the present study managed to measure both the implanted dexamethasone and the endogenous 1α-OHB at the same time. After 7 days, the synthetic dexamethasone showed plasma concentrations three orders higher than those of 1α-OHB in *S. canicula* [19] or cortisol in teleosts [63]. This situation may trigger pharmacological responses in the studied species, so it will be necessary to be cautious regarding the interpretation of our results. However, the effect of dexamethasone in *S. canicula* is to decrease circulating concentration of 1α-OHB suggesting a negative feed-back of the hypothalamic–hypophyseal–interrenal axis, as expected from an analogous hormone that competes for the same receptors [1]. Our results coincide with the described effect of dexamethasone after 4 days in *S. canicula*, where 1α-OHB concentration decreased to half of its basal values [17].

The study from Hazon and Henderson, altogether with others, highlighted that 1α-OHB triggers MC responses in sharks [64,65]. The present study also seems to evidence MC responses through the increase in muscle water content due to dexamethasone treatment, though further studies are necessary to fully elucidate them (it was not the goal of this study). It was previously described in *S. canicula* that acute-stress responses (with high levels of plasma 1α-OHB) induced osmoregulatory imbalances, including changes in muscle water content [19,38,39]. Moreover, chronic administration of dexamethasone in teleost fish is related to both MC and GC changes [23]. Altogether, the use of dexamethasone to study the putative corticoid roles of corticosteroids (MC and GC) in sharks seems to be a valid approach.

### 4.2. Glycolytic Effects of Long-Term Dexamethasone Treatment in Catsharks

In this study, the lack of changes in the basal physiological components for homeostasis (plasma hematocrit and pH, as well as glucose and lactate levels in plasma, liver, and white muscle) may indicate a possible homeostatic balance achieved during acclimation to the implanted dexamethasone [66]. As there are no previous studies conducted in elasmobranchs, acclimation time after a chronic situation is unknown. However, the equilibrium achieved in the present study coincided with that described in teleost fish, which occurs one week after the onset of the stress situation [31,67]. Thus, slow-release implants of dexamethasone induced an allostatic state in *S. canicula* during the first 7 days, and all changes described may be associated to a chronic situation. However, it would be necessary to confirm this by analyzing the time course of physiological changes in this species, which was not included in the present study to minimize the number of animals employed.

Here, it is highlighted that dexamethasone (presumably through GR) affected glycogen stores differentially (liver and white muscle) and hence, carbohydrate metabolism (including glycolysis, and gluconeogenesis pathways) during a long-term treatment, coinciding with a previous study conducted in a teleost fish. Specifically, *S. aurata* chronically fed with dexamethasone showed a higher HSI than the control group due to hepatic glycogen accumulation, while glycogenolysis was enhanced in white muscle of cortisol-fed fish [23]. While liver is considered the principal organ of glucose homeostasis in vertebrates, glycogen is also stored in the muscle, where glucose phosphorylation capacity is higher than in the hepatocytes [32]. Under chronic exposure to dexamethasone, *S. canicula* stored glycogen in the liver while white muscle consumed it. Thus, glycolysis and gluconeogenesis pathways are also stimulated in the latter tissue, as seen by the activity of the enzymes HK, PK, and LDH. These results indicate that the increased energy needs during long-term corticosteroid treatment in sharks could be supported by consumption of carbohydrates in white muscle, as described before in the teleost *Solea senegalensis* [67]. However, due to the importance of ketone bodies in the metabolism of sharks [22], it will be interesting to assess the contribution of these metabolites to satisfy the energy demand imposed by dexamethasone implants. Moreover, the observed changes in GP activity, reduced in the liver and enhanced in the muscle, supported the results obtained in the present study. In this line, it was described that elasmobranchs have two isozymes of glycogen phosphorylase: a liver and a muscle form [68], which may help to understand the differences between both tissues during chronic dexamethasone treatment. Further studies assessing differential changes in these isoforms are necessary to test this hypothesis.

### 4.3. Acute Ex Vivo Glucocorticoid Actions of Corticosteroids

To describe the acute responses to corticosteroids in *S. canicula*, this study highlights the usefulness of ex vivo tissue culture. The tissues employed herein are shown to be alive and functional during the 5 h tested since basal glucose secretion does not vary in control groups, but it is stimulated in the presence of GCs. Thus, liver and muscle explants, incubated under physiological conditions and exposed to non-natural corticosteroids in sharks (cortisol and dexamethasone, different to the native 1α-OHB of sharks), are able to secrete glucose for up to 5 h as a paradigmatic GC response. This study shows that cortisol and dexamethasone can exploit the promiscuity of the 1α-OHB GC receptor, which has been previously demonstrated in the little skate [12] and in the elephant shark [50]. However, glucose secretion rate diminishes through time, with maximum values within the first 1 h incubation. This time coincides with maximum blood glucose concentration in teleosts after an acute-stress situation such as air exposure [20,52], and may be related to the described effect of dexamethasone in ex vivo cultures of rat hepatocytes, which activated glycogen phosphorylase activity after 10 min [69]. However, sharks and other elasmobranchs showed delayed in vivo hyperglycemia following acute challenges [19,26,70,71], while some other authors described the absence of changes [21,72,73,74,75] that may be related to the low levels of plasma glucose described in this taxa [32]. Altogether, if glucose production cannot match the rise in its consumption after enhancement of anaerobic metabolism following acute-stress processes, hyperglycemia would not be observed in vivo, reinforcing the idea of ex vivo approaches like the one described in this study to gain further knowledge on this topic.

When incubating both liver and muscle explants of *S. canicula* under a wide range of cortisol and dexamethasone concentrations (from 3 nM, physiological levels of 1α-OHB described in catsharks after acute air-exposure [19], to 30 µM that can be considered pharmacological levels), glucose secretion rate increased in both tissues, as expected action of GC hormones. Notwithstanding the fact that we employed two corticosteroids that are not naturally present in sharks in these tests, there are no differences in the glucose secretion rates at the same concentration of both hormones. Under these circumstances, maximum glucose secretion was observed at the described maximum physiological concentration of 3 nM 1α-OHB in *S. canicula* after an acute-stress [19]. This result may indicate that the active site of the putative GR recognizes cortisol and dexamethasone equally (or at least with similar affinities), coinciding with the described low-sensitivity of elasmobranch GRs activated by multiple corticosteroids [50,76]. The possible affinity of the 1α-OHB GR for other corticosteroids not naturally present in *S. canicula* may certainly be describing physiological processes that occur in vivo in sharks in the presence of cortisol and dexamethasone. The described lack of differences in glucose secretion at corticosteroid concentrations above 3 nM may indicate the existence of only one GR in this species—as described in Chondrichthyes—which have differentiated GR and MR after duplication of the whole genome [11,12]. However, while this glucose secretion enhancement may be clearly associated to one GR in the liver, responses in white muscle seem to show something slightly different, with a decrease in the secretion rate at 30 nM corticosteroids, returning to the maximum rates reached at 3 nM above 300 nM. This response may be associated with the presence of two GR-like in catsharks, which was already described in different tissues, including muscle, in teleosts [77,78]. However, further studies including more ex vivo experiments with purified 1α-OHB tested at its physiological range in sharks are needed. Moreover, describing possible GR isoforms results necessary, also evaluating if the MR of sharks has a dual role as GR. On the other side, the observed changes in glucose secretion in muscle explants could be ascribed to an artifact product of the sensitivity of the measurement method. In this sense, maximum glucose secretion rates are three times higher in liver than in muscle in the present study. For this reason, we chose the liver for further studies related to glucose secretion, coinciding with the described importance of this tissue to fuel with glucose energy demands imposed by primary stress responses (including corticosteroid actions) in fish [27,79].

Previous studies related to ex vivo effects of corticosteroids in fish were conducted in teleost hepatocytes, gills, and kidney, evaluating changes in RNA transcription of different genes [54,58,78,80]. However, to the best of our knowledge, none of them evaluated the secretion of glucose as a benchmark of GC activity. Furthermore, to our knowledge, the fact that the initial glucose concentration in the incubation medium is affecting the ability to secrete this metabolite from the liver has never been described in vertebrates. In this sense, maximum secretion rates occurred at physiological basal levels of glucose in the incubation medium (2.3 mM glucose), while they decreased drastically in the absence of external glucose, and to a lesser extent when glucose incubation medium concentration doubled (4.6 mM glucose). Further studies are mandatory to unravel this question, including analysis of transmembrane glucose transporters [34,81], the feasible action of non-canonical corticosteroid mechanisms [35,37] that may be associated to membrane processes, or the presence of glucosensors involved in GC actions.

### 4.4. Glucocorticoid Actions Mediated by GR in Catsharks

Finally, the main finding of this study is the involvement of GR in the GC response to non-naturally occurring corticosteroids. By using the GR inhibitor mifepristone (or RU-486)—previously tested in ex vivo cultures of murine mammary carcinomas, human endometrial cells, salmon gills, and trout hepatocytes [54,56,57,58]—we describe, for the first time, the direct involvement of corticosteroids (cortisol and dexamethasone) as GC hormones in sharks. This is in accordance with a recent study describing the in vitro transcriptional activation of both GR and MR of elephant shark at a concentration of 10 nM cortisol (EC50 = 35 nM), where RU-486 served as specific inhibitor of the GR [50]. Mifepristone completely inhibited corticosteroids-induced glucose secretion in liver explants of *S. canicula* using both cortisol and dexamethasone as putative hormones similar to the endogenous corticosteroid in sharks. Therefore, although further studies are necessary, it is reasonable to think that 1α-OHB has a GC action in sharks, as previously stated [19]. Further studies should focus on the mechanisms of action of this hormone in other tissues, looking at the implications at the systemic level. Something else to keep in mind is that mifepristone also inhibits progesterone receptors [48], so it should be demonstrated that these are not related to the release of glucose mediated by cortisol or dexamethasone in this species.

## 5. Conclusions

This study demonstrated that cortisol and dexamethasone, corticosteroids not present in sharks, have glucocorticoid actions in catsharks. They mobilize glucose from storage tissues (liver and white muscle) rapidly, with maximum secretion rates occurring within the first hour after the onset of the peak in circulating hormones. These actions, mediated by mifepristone-inhibited receptors (mainly GR, but could also be progesterone receptors), coincide with those described in teleosts and terrestrial vertebrates. The implications of this study are of paramount importance to understand the mechanisms of action of the intermediary metabolism, and the stress responses, in ancient vertebrates.

## Figures and Tables

**Figure 1 biology-11-00056-f001:**
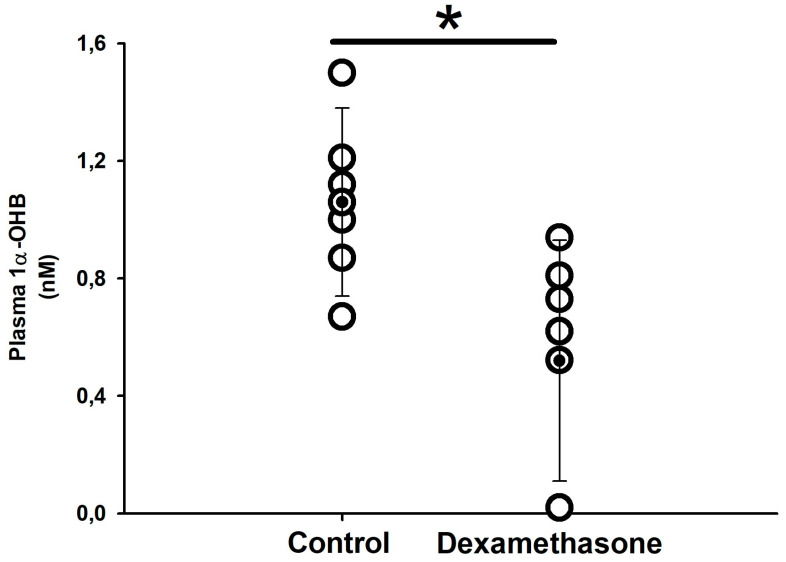
Plasma 1α-OHB in catsharks (*S. canicula*) 7 days after IP-injection of control or dexamethasone (Dex) slow-release implants. Data are expressed as single plots per individual. Mean and SD are shown as black dots and vertical lines, respectively. The asterisk (*) indicates significant differences between both groups (*p* < 0.05, Student´s *t*-test, n = 7).

**Figure 2 biology-11-00056-f002:**
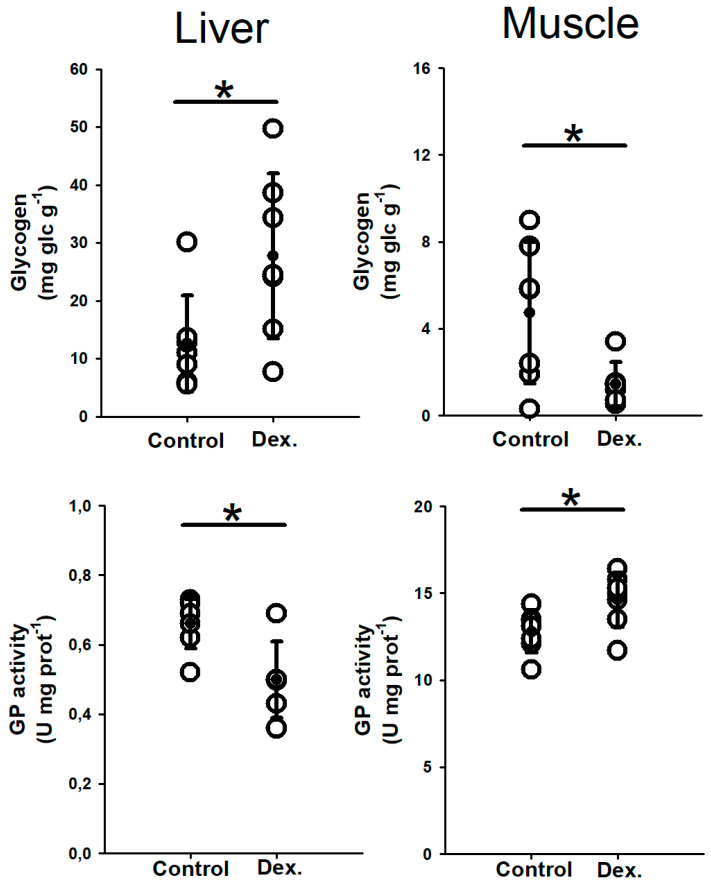
Stored glycogen (mg glucidic units g^−1^) and glycogen phosphorylase (GP) activity (U mg prot^−1^) in liver and white muscle of catsharks after 7 days of corticosteroid treatment. Mean and SD are shown as black dots and vertical lines, respectively. The asterisk (*) indicates significant differences between both groups (*p* < 0.05, Student´s *t*-test, n = 7).

**Figure 3 biology-11-00056-f003:**
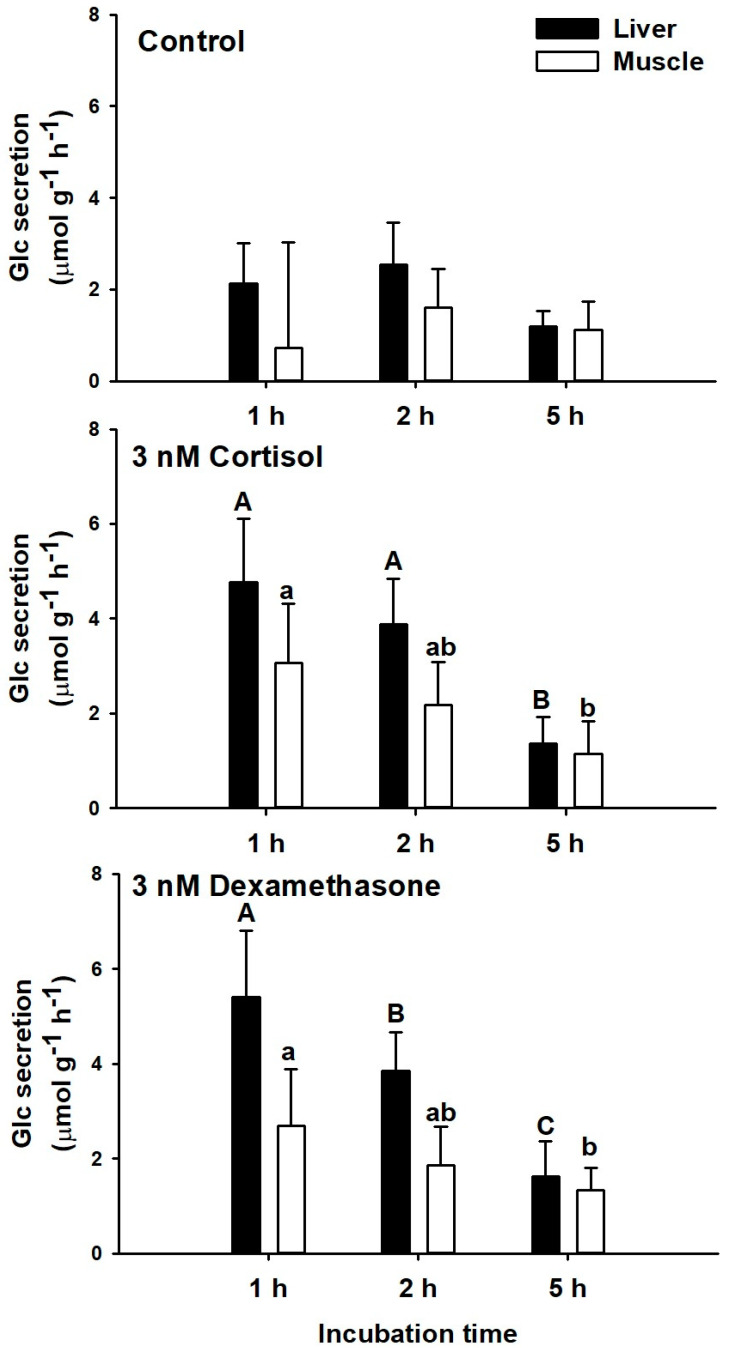
Glucose secretion rates (µmol g^−1^ h^−1^) in liver and muscle of *S. canicula* after 1, 2, and 5 h of incubation with DMSO alone (control group), 3 nM cortisol, or 3 nM dexamethasone. Glucose secretion at each time is represented by black bars (liver) or white bars (muscle). Data are expressed as mean ± SD. Different capital and lowercase letters indicate significant differences with time for the liver and muscle, respectively (*p* < 0.05, one-way ANOVA followed by a Tukey´s post hoc test, n = 6).

**Figure 4 biology-11-00056-f004:**
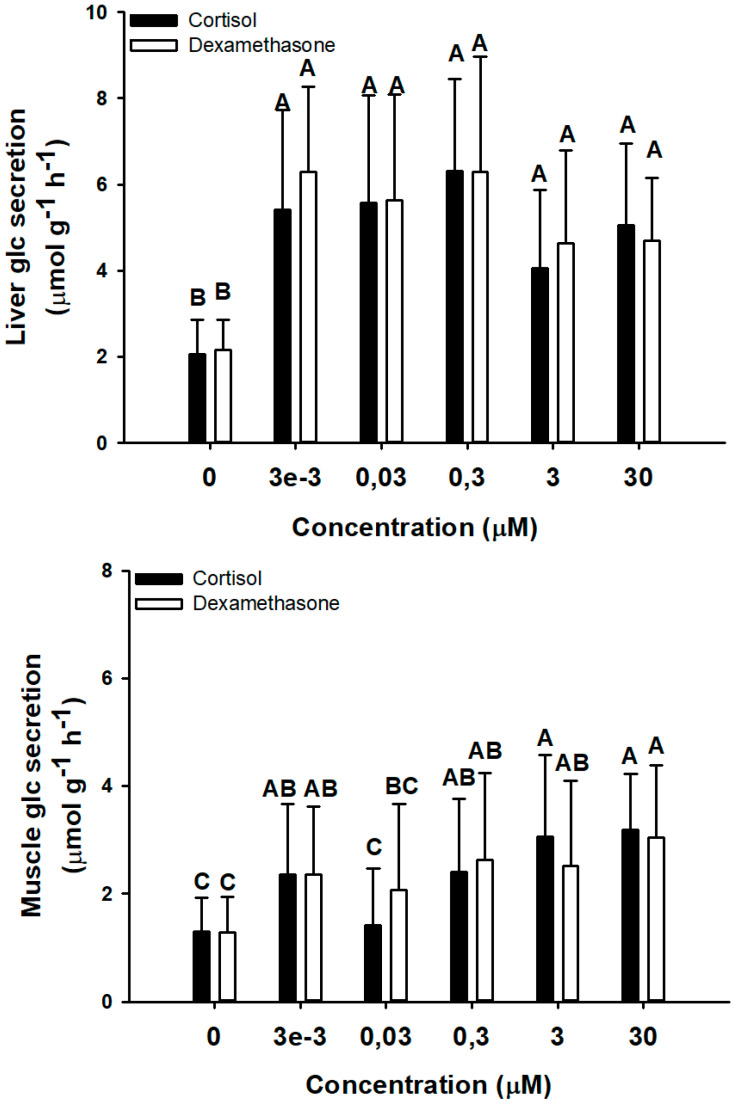
Glucose secretion rates (µmol g^−1^ h^−1^) in liver and muscle of *S. canicula* after 1 h ex vivo incubation under a range of cortisol (black bars) and dexamethasone (white bars) concentrations (0, 0.003, 0.03, 0.3, 3, and 30 µM). Data are expressed as mean ± SD. Different capital letters indicate significantly different groups (*p* < 0.05, two-way ANOVA followed by a Tukey´s post hoc test, n = 16 explants per tissue, from 2 adult females).

**Figure 5 biology-11-00056-f005:**
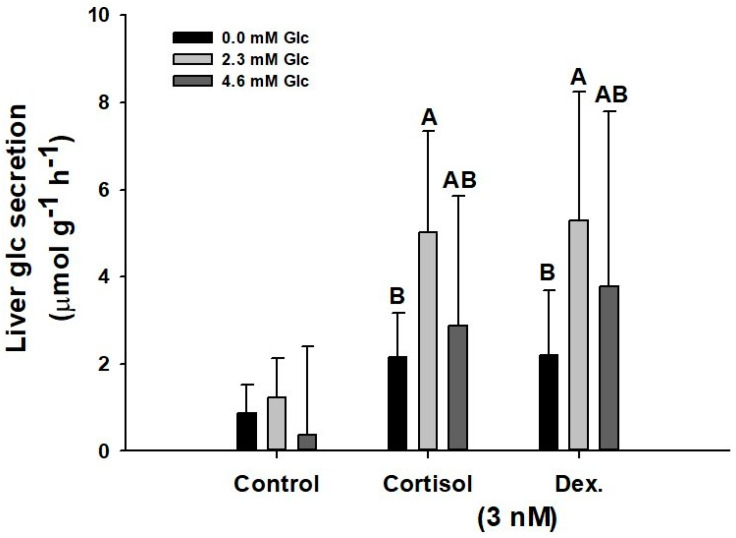
Hepatic glucose secretion rates (µmol g^−1^ h^−1^) after 1 h ex vivo incubation with 3 nM cortisol or dexamethasone and different initial glucose concentrations in the incubation medium (0.0, 2.3, or 4.6 mM) in *S. canicula*. Data are expressed as mean ± SD. Different capital letters indicate significant differences due to initial glucose concentration for each hormone (*p* < 0.05, two-way ANOVA followed by a Tukey´s post hoc test, n = 6).

**Figure 6 biology-11-00056-f006:**
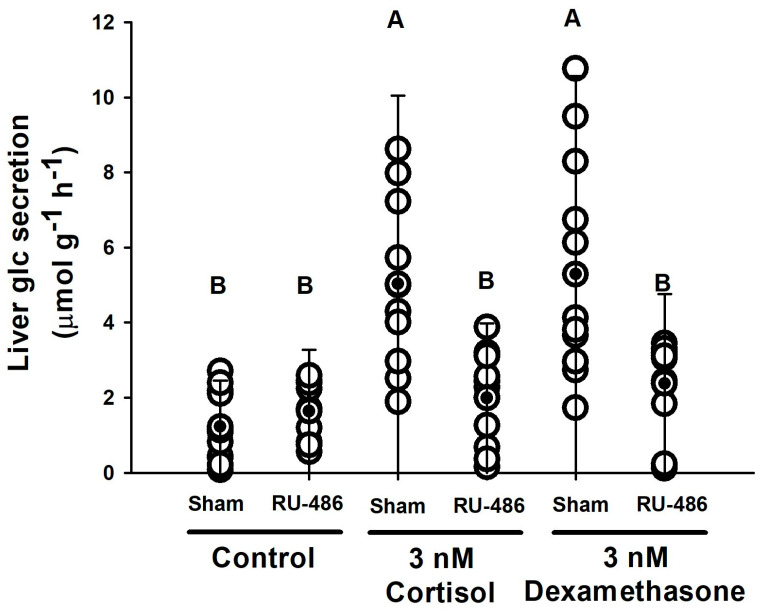
Glucose secretion rates (µmol g^−1^ h^−1^) in liver explants of *S. canicula* incubated with 3 nM cortisol or dexamethasone and a glucocorticoid specific inhibitor (30 µM RU-486 or mifepristone, white bars). The sham-control group was incubated in the presence of DMSO as the solvent necessary to pre-dissolve these compounds. Each dot represents a single explant (two liver explants, duplicates, per animal). Mean and SD are shown as black dots and vertical lines, respectively. Different capital letters indicate significantly different groups (*p* < 0.05, two-way ANOVA followed by a Tukey´s post hoc test, n = 6 animals).

**Table 1 biology-11-00056-t001:** Changes in plasma, liver, and muscle parameters due to long-term (7 days) dexamethasone IP-implants in catsharks (*S. canicula*). Data are expressed as mean ± SD. Asterisks (*) indicate significant differences between both groups (*p* < 0.05, Student´s *t*-test, n = 7). The activity (act.) of HK (hexokinase), PK (pyruvate kinase), LDH (lactate dehydrogenase), and FBP (fructose bi-phosphatase) were analyzed. HSI = hepatosomatic index.

Tissue	Parameter	Control	Dexamethasone	*p*-Value
Plasma	Haematocrit (%)	18.1 ± 3.3	17.3 ± 2.5	0.593
	pH	7.69 ± 0.05	7.66 ± 0.05	0.231
	Glucose (mM)	1.91 ± 0.60	2.31 ± 1.10	0.421
	Lactate (mM)	0.85 ± 0.71	0.72 ± 0.36	0.683
Liver	HSI	3.26 ± 0.78	3.70 ± 1.09	0.401
	Glucose (mg g^−1^)	2.65 ± 0.69	2.45 ± 0.59	0.568
	Lactate (mg g^−1^)	1.39 ± 1.22	1.19 ± 0.52	0.697
	HK act. (U mg prot^−1^)	0.08 ± 0.04	0.07 ± 0.03	0.614
	PK act. (U mg prot^−1^)	3.20 ± 0.71	3.28 ± 0.75	0.354
	LDH act. (U mg prot^−1^)	0.02 ± 0.02	0.01 ± 0.02	0.119
	FBP act. (U mg prot^−1^)	0.90 ± 0.16	0.97 ± 0.15	0.437
Muscle	Water content (%)	68.7 ± 2.63	73.0 ± 2.27 *	0.006
	Glucose (mg g^−1^)	3.13 ± 1.3	2.88 ± 0.99	0.704
	Lactate (mg g^−1^)	36.1 ± 7.78	42.0 ± 3.80	0.099
	HK act. (U mg prot^−1^)	0.17 ± 0.06	0.33 ± 0.15 *	0.019
	PK act. (U mg prot^−1^)	71.4 ± 17.32	108.9 ± 10.45 *	0.000
	LDH act. (U mg prot^−1^)	0.37 ± 0.11	0.60 ± 0.17 *	0.014
	FBP act. (U mg prot^−1^)	0.39 ± 0.20	0.29 ± 0.12	0.304

## Data Availability

The data presented in this study are available on request from the corresponding author.

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
