# Peer review of "Cortisol and Dexamethasone Mediate Glucocorticoid Actions in the Lesser Spotted Catshark (Scyliorhinus canicula)"

_biology, 2021, doi:10.3390/biology11010056_

Round 1
Reviewer 1 Report
Cabrera‐Busto et al. assessed whether cortisol and dexamethasone mediate glucocorticoid actions in the lesser spotted catshark. GC effects of Cortisol and dexamethasone have been well-established in most vertebrates. The authors showed these same effects on catshark. On the other hand, they (for obvious reasons) did not assess the function of the main corticosteroid in these species, the 1α‐OHB. In fact, the authors merely showed that corticosteroids not naturally present in sharks may exhibit glucocorticoid effects in this species, which is I believe something that does not need clarification, as exon length and amino acid sequence of GC receptors is conserved among all classes of vertebrate (highly conserved). In my opinion, it would be much more beneficial if the authors sought to find the hormone that exhibits main GC effects in lesser spotted catshark, as that is the actual curiosity.
Minor comments:
- mean and SEM is inappropriate way of data presentation. Please use mean and SD, as SEM is not indicator of data dispersion.
- The first part of the Figure 1 should be omitted.
- Adjust the abstract section in such manner that it is obvious when cortisol is used.
Author Response
REVIEWER 1
Comments and Suggestions for Authors
Cabrera‐Busto et al. assessed whether cortisol and dexamethasone mediate glucocorticoid actions in the lesser spotted catshark. GC effects of Cortisol and dexamethasone have been well-established in most vertebrates. The authors showed these same effects on catshark. On the other hand, they (for obvious reasons) did not assess the function of the main corticosteroid in these species, the 1α‐OHB. In fact, the authors merely showed that corticosteroids not naturally present in sharks may exhibit glucocorticoid effects in this species, which is I believe something that does not need clarification, as exon length and amino acid sequence of GC receptors is conserved among all classes of vertebrate (highly conserved). In my opinion, it would be much more beneficial if the authors sought to find the hormone that exhibits main GC effects in lesser spotted catshark, as that is the actual curiosity.
Answer: The Reviewer is correct about the suitability of having the 1α-OHB available to know its real effects. In the absence of it, we consider that the present study may be useful for future approaches to unravel its GC effects. We appreciate all comments and suggestions.
Minor comments:
- mean and SEM is inappropriate way of data presentation. Please use mean and SD, as SEM is not indicator of data dispersion.
Answer: This is a subject that generates debate at times, and we will try to explain the reasons why we have decided to use SEM instead of SD. In case the Reviewer continues to consider exchanging SEM for SD, we will of course do what it is proposed. We chose SEM because it not only considers the dispersion of the data (as does the SD, since it requires the SD for its calculation, SEM = SD / SQRT (N)), but also the number of samples, which greatly limits the effect of the experimental group, being more sensitive to changes due to individual variation. If the Reviewer still considers SD to be more appropriate, we will include it in place of SEM.
- The first part of the Figure 1 should be omitted.
Answer: What is the reviewer referring to by removing the first part? Deleting the figure of “plasma dexamethasone”? If that is the case, we therefore show in Figure 1 just the concentration of plasma 1α-OHB.
- Adjust the abstract section in such manner that it is obvious when cortisol is used.
Answer: We have modified the penultimate sentence to make it clear where cortisol has been used. Now it is read as “Dose-response curves induced by cortisol and DEX, altogether with the use of specific GR inhibitor mifepristone, confirmed the involvement of GR mediating glucose secretion”.
Reviewer 2 Report
In my opinion, the manuscript entitled “Cortisol and dexamethasone mediate glucocorticoid actions in the lesser spotted catshark (Scyliorhinus canicula)” is well designed and performed. The scientific background is sufficient, but there are some minor concerns about the presentation, which should be addressed.
Line 77-78: Revise the sentence.
Line 86-87: Revise the sentence.
Line 106: S. Canicula
Line 122: Revise the sentence.
Line 129-132: Revise the sentence.
Line 331-333: Revise the sentence.
Line 340: “S. canicular”: Must be italic.
Line 350: The full name for abbreviations must be provided when they first appear in the text. Please, check them throughout the text. Their full names should also be listed below the table.
Line 413-414: Revise the sentence.
Author Response
REVIEWER 2
Comments and Suggestions for Authors
In my opinion, the manuscript entitled “Cortisol and dexamethasone mediate glucocorticoid actions in the lesser spotted catshark (Scyliorhinus canicula)” is well designed and performed. The scientific background is sufficient, but there are some minor concerns about the presentation, which should be addressed.
Answer: We appreciate all Reviewer´s comments and proceed to include the suggestions in the text.
Line 77-78: Revise the sentence.
Answer: We change “slowing down” by “compromising”.
Line 86-87: Revise the sentence.
Answer: The sentence has been modified and now it reads as “Glucose metabolism also results of major importance after short-term situations as described in vertebrates, including sharks, to provide energy to demanding tissues”.
Line 106: S. Canicula
Answer: We appreciate the sharp revision of the Reviewer and modified the word as suggested.
Line 122: Revise the sentence.
Answer: The sentence has been modified and now it reads as “Due to the absence of purified 1α-OHB, alternatives have been sought that may provide some information on the effects of corticosteroids on this species”.
Line 129-132: Revise the sentence.
Answer: The sentence has been modified and now it reads as “The amount of implanted dexamethasone was based on a previous study carried out in S. canicula, where daily injections were administered for 4 days, but also on a study carried out in the teleost S. aurata treated with slow-release cortisol implants”.
Line 331-333: Revise the sentence.
Answer: The sentence has been modified and now it reads as “Plasma 1α-OHB concentration decreased by half in the dexamethasone-treated group (0.52 ± 0.17 nM) compared to the control group 1.06 ± 0.14 nM”.
Line 340: “S. canicular”: Must be italic.
Answer: The error has been corrected thanks to the comment.
Line 350: The full name for abbreviations must be provided when they first appear in the text. Please, check them throughout the text. Their full names should also be listed below the table.
Answer: We appreciate the comment and thoroughly revised the abbreviations along the text. As for the case of the tables, the full names of the enzymes are written in their headings.
Line 413-414: Revise the sentence.
Answer: The sentence has been modified and now it reads as “To test the ex vivo glucose secretion rates by the tissues (liver and muscle), we considered necessary to test the effects of the initial glucose concentration in the incubation medium”.
Reviewer 3 Report
The authors present evidence that cortisone and dexamethasone affect glucose homeostasis in the lesser spotted catshark. While this manuscript is very limited in the amount of data presented, it is understandable given the amount of work required to obtain the animals needed. The studies are also done well, and the manuscript is clearly written, so I believe this article is worthy of publication once the following concerns have been addressed.
Reviewer concerns:
Table 1. It would be helpful if the authors provided the p-value for each comparison in the table. In addition, do these numbers have to be corrected for repeated measures?
General point. All figures should include markers for the individual biological replicates so the reader can evaluate the variation in each group of samples.
Figure 3. It would be helpful if the authors showed data for control explants to help the reader appreciate how time in culture affects glucose secretion. While the authors state that glucose secretion rates stayed constant in explants over the five-hour period (line 371), they only show one value in their description.
Figure 4. The figure legend states that there are only 2 samples per group. If that is the case, how did the authors perform a two-way ANOVA since there aren’t enough samples to determine variance?
Figure 6. RU-486 also affects progesterone signaling and may thus have no
The authors present evidence that cortisone and dexamethasone affect glucose homeostasis in the lesser spotted catshark. The studies are done well, and the manuscript is clearly written so I believe this article is worthy of publication. However, I should say that this paper is rather limited in scope and might be better suited to a more specialized journal
I have concerns that should be addressed prior to publication.
Reviewer concerns:
Table 1. It would be helpful if the authors provided the p-value for each comparison in the table. In addition, do these numbers have to be corrected for repeated measures?
General point. All figures should include markers for the individual biological replicates so the reader can evaluate the variation in each group of samples.
Figure 3. It would be helpful if the authors showed data for control explants to help the reader appreciate how time in culture affects glucose secretion. While the authors state that glucose secretion rates stayed constant in explants over the five-hour period (line 371), they only show one value in their description.
Figure 4. The figure legend states that there are only 2 samples per group. If that is the case, how did the authors perform a two-way ANOVA since there aren’t enough samples to determine variance?
Figure 6. RU-486 also affects progesterone signaling and may thus have non-specific effects. While RU-486 is the only glucocorticoid receptor antagonist available, there are inhibitors of the progesterone receptor that the authors could use as a negative control. This would help ensure that the effects the authors are seeing with RU-486 are specific to glucocorticoid signaling.
n-specific effects. While RU-486 is the only glucocorticoid receptor antagonist available, there are inhibitors of the progesterone receptor that the authors could use as a negative control. This would help ensure that the effects the authors are seeing with RU-486 are specific to glucocorticoid signaling.
Author Response
REVIEWER 3
Comments and Suggestions for Authors
The authors present evidence that cortisone and dexamethasone affect glucose homeostasis in the lesser spotted catshark. While this manuscript is very limited in the amount of data presented, it is understandable given the amount of work required to obtain the animals needed. The studies are also done well, and the manuscript is clearly written, so I believe this article is worthy of publication once the following concerns have been addressed.
Answer: We appreciate the Reviewer´s comments and proceed to apply the suggestions in the text.
Reviewer concerns:
Table 1. It would be helpful if the authors provided the p-value for each comparison in the table. In addition, do these numbers have to be corrected for repeated measures?
Answer: The p-values have been included in the table. We do not consider appropriate to employ repeated measures in the in vivo experiment, since each animal was sampled only once.
General point. All figures should include markers for the individual biological replicates so the reader can evaluate the variation in each group of samples.
Answer: As suggested, we have made the modifications to the figures, showing the individual points of each animal / sample. However, we have only done it in those figures that we consider that they can get richer thanks to these changes (figures 1, 2 and 6).
Figure 3. It would be helpful if the authors showed data for control explants to help the reader appreciate how time in culture affects glucose secretion. While the authors state that glucose secretion rates stayed constant in explants over the five-hour period (line 371), they only show one value in their description.
Answer: We have included in Figure 3 the results from the control groups.
Figure 4. The figure legend states that there are only 2 samples per group. If that is the case, how did the authors perform a two-way ANOVA since there aren’t enough samples to determine variance?
Answer: This figure was done with 8 explants per animal, though the number of animals was 2 (n = 2). We did not want this figure to be more important due to the low number of replicas (individuals). The figure legend has been modified to clarify the statistical procedure.
Figure 6. RU-486 also affects progesterone signaling and may thus have non-specific effects. While RU-486 is the only glucocorticoid receptor antagonist available, there are inhibitors of the progesterone receptor that the authors could use as a negative control. This would help ensure that the effects the authors are seeing with RU-486 are specific to glucocorticoid signaling.
Answer: The Reviewer is right, mifepristone also inhibits progesterone receptors. However, we have not found any study that relates the action of progesterone receptors with glucose secretion directly. We have included this information, along with supporting references, in the text.
Reviewer 4 Report
The data presented in the manuscript 'Cortisol and dexamethasone mediate glucocorticoid actions in the lesser spotted catshark' are compelling and novel. These data strongly suggest that there is intact glucocorticoid-glucocorticoid receptor (GC-GR) signaling that regulates glucose homeostasis in the species. My main concern is that the conclusions drawn from these data are overstated as GR signaling was not directly assessed. The authors utilized RU486 to determine GC/GR activity and state it is a GR specific antagonist, but it is well-know to have high affinity for the progesterone receptor and it appears that PR is also expressed in this species. I realize the glucose data along with the use of DEX and the RU486 experiment support GC-GR activity, nevertheless direct activity should be assessed prior to making these statements. It is my suggestion that the authors edit the text to reflect this, or provide additional experiments to assess GC-GR activity in these tissues directly such as ChIP, GR knockdowns or at least measure transcriptional changes in GR-regulated genes with known GREs (such as liver gluconeogenic genes).
Author Response
REVIEWER 4
Comments and Suggestions for Authors
The data presented in the manuscript 'Cortisol and dexamethasone mediate glucocorticoid actions in the lesser spotted catshark' are compelling and novel. These data strongly suggest that there is intact glucocorticoid-glucocorticoid receptor (GC-GR) signaling that regulates glucose homeostasis in the species. My main concern is that the conclusions drawn from these data are overstated as GR signaling was not directly assessed. The authors utilized RU486 to determine GC/GR activity and state it is a GR specific antagonist, but it is well-know to have high affinity for the progesterone receptor and it appears that PR is also expressed in this species. I realize the glucose data along with the use of DEX and the RU486 experiment support GC-GR activity, nevertheless direct activity should be assessed prior to making these statements. It is my suggestion that the authors edit the text to reflect this, or provide additional experiments to assess GC-GR activity in these tissues directly such as ChIP, GR knockdowns or at least measure transcriptional changes in GR-regulated genes with known GREs (such as liver gluconeogenic genes).
Answer: We appreciate the Reviewer´s comments and are aware of the limitations of our study. As mifepristone also inhibits progesterone receptors, some cross-reactivity may occur when using this inhibitor. However, we have not found any study that relates the action of progesterone receptors with glucocorticoid activity. We have included this information, along with supporting references, in the text. Moreover, the conclusions have been also modified to accommodate this suggestion and now it reads as: “These actions, mediated by mifepristone-inhibited receptors (mainly GR, but could also be progesterone receptors), coincide with those described in teleosts and terrestrial vertebrates”.
Round 2
Reviewer 1 Report
Dear authors, the SEM is a measure of precision for an estimated population mean. Unlike SD, SEM is not a descriptive statistics and should not be used as such. However, many authors incorrectly use the SEM as a descriptive statistics to summarize the variability in their data because it is less than the SD, implying incorrectly that their measurements are more precise. The SEM is correctly used only to indicate the precision of estimated mean of population. Even then however, a 95% confidence interval should be preferred. Therefore, I strongly advise to change SEM to SD.
The authors changed Figure 1 appropriately.
Author Response
REVIEWER 1
Comments and Suggestions for Authors
Dear authors, the SEM is a measure of precision for an estimated population mean. Unlike SD, SEM is not a descriptive statistics and should not be used as such. However, many authors incorrectly use the SEM as a descriptive statistics to summarize the variability in their data because it is less than the SD, implying incorrectly that their measurements are more precise. The SEM is correctly used only to indicate the precision of estimated mean of population. Even then however, a 95% confidence interval should be preferred. Therefore, I strongly advise to change SEM to SD.
Answer: We exchange SEM by SD according to the Reviewer’s suggestion. Moreover, we appreciate the explanation, as it will be of use in the future.
The authors changed Figure 1 appropriately.
Reviewer 3 Report
I very much appreciate the authors showing their individual data points, but I think they should add lines representing the mean and standard deviation (or SEM if preferred). Beyond that, the manuscript looks fine.
Author Response
Comments and Suggestions for Authors
I very much appreciate the authors showing their individual data points, but I think they should add lines representing the mean and standard deviation (or SEM if preferred). Beyond that, the manuscript looks fine.
Answer: We really appreciate the comment. Mean and SD were also included in the figures.
Reviewer 4 Report
I appreciate the authors addressing my concerns regarding the possible involvement of PR in these findings. There are still a few minor grammatical errors throughout that need to be corrected, such as in line 614 "
unless it is still to be demonstrated" should be rephrased.
Author Response
REVIEWER 4
Comments and Suggestions for Authors
I appreciate the authors addressing my concerns regarding the possible involvement of PR in these findings. There are still a few minor grammatical errors throughout that need to be corrected, such as in line 614 "unless it is still to be demonstrated" should be rephrased.
Answer: We want to thank this Reviewer for the helpful advices. The manuscript has been now revised by an English native speaker and, hopefully, all grammatical errors are corrected.